# Fall Detection Algorithm Based on Inertial Sensor and Hierarchical Decision

**DOI:** 10.3390/s23010107

**Published:** 2022-12-22

**Authors:** Liang Zheng, Jie Zhao, Fangjie Dong, Zhiyong Huang, Daidi Zhong

**Affiliations:** 1Bioengineering College, Chongqing University, Chongqing 400044, China; 2The 15th Research Institute of China Electronics Technology Group Corporation, Beijing 100083, China; 3Wuhan Branch of Beijing Zunguan Technology Co., Ltd., Wuhan 430079, China; 4School of Microelectronics and Communication Engineering, Chongqing University, Chongqing 400044, China

**Keywords:** fall detection, feature dimensionality reduction, XGBoost

## Abstract

With the aging of the human body and the reduction in its physiological capacities, falls have become a huge threat to individuals’ physical and mental health, leading to serious bodily damage to the elderly and financial pressure on their families. As a result, it is vital to design a fall detection algorithm that monitors the state of human activity. This work designs a human fall detection algorithm based on hierarchical decision making. First, this work proposes a dimensionality reduction approach based on feature importance analysis (FIA), which optimizes the feature space via feature importance. This procedure reduces the dimension of features greatly and reduces the time spent by the model in the training phase. Second, this work proposes a hierarchical decision-making algorithm with an XGBoost model. The algorithm is divided into three levels. The first level uses the threshold approach to make a preliminary assessment of the data and only transfers the fall type data to the next level. The second level is an XGBoost-based classification algorithm to analyze again the type of data which remained from the first level. The third level employs a comparison method to determine the direction of the falling. Finally, the fall detection algorithm proposed in this paper has an accuracy of 98.19%, a sensitivity of 97.50%, and a specificity of 98.63%. The classification accuracy of the fall direction reaches 93.44%, and the algorithm can efficiently determine the fall direction.

## 1. Introduction

Population ageing is a common phenomenon in many countries due to declining fertility and increasing life expectancy [1]. As of November 2020, China’s population which were aged 65 and above totaled 190.64 million, accounting for 13.5% of the total population. As can be seen from Figure 1, the proportion of China’s population aged 65 and over reached 7% in 2000, indicating that China has entered an aging society. The proportion of the population aged 65 and above and the growth rate of its size continues to accelerate. Today, China is only one step away from becoming a deeply aging society. Population aging is a major victory for human beings in many aspects such as medical technology and the environment and technology, but it is also a challenge to society [1]. The increase in the degree of aging is not only reflected in the changes in the elderly population itself, but also in the decline of the working-age population and the aging of the labor force [2]. Changes in the structure of the population will further aggravate the problems of the elderly, and the problem of falls and the phenomenon of empty nesters will become increasingly prominent. The literature [3] shows that falls are the fourth leading cause of injury-related death in China, and it ranks first among those over 65 years old.

The elderly are within a high-risk group regarding the risk of falls. Once a fall occurs, there will be a high rate of hospitalization and casualty, which will bring great financial burdens to the family. If the treatment is not timely, it will directly lead to diseases such as fractures and paralysis, and it may also indirectly induce other diseases [4]. As falls have increasingly become a key factor threatening the quality of life and safety of the elderly, the prevention and protection of the elderly from falls has attracted more and more attention [5]. In view of the great suddenness and randomness of the occurrence of falls, the detection of falls has become the focus of much research.

Wearable devices are widely used in many fields, including fall detection. Reference [6] achieves the detection of a fall by setting thresholds for resultant acceleration, the angle, and the angular velocity. The study was tested using the fall dataset named Sisfall. Due to the loose threshold conditions, the accuracy rate reaches 90.3% and the sensitivity reaches 100%, but the highest specificity only reaches 83.9%. These experimental results also show that the algorithm has a 100% false positive rate for behaviors such as lying down and sitting down, indicating that the algorithm cannot distinguish between high acceleration behaviors and falls.

The advantage of this work is that the machine learning method is used instead of the threshold method, which can effectively reduce the probability of false alarms and obtain a higher generalization ability and reliability. At the same time, this work uses public datasets to make the results comparable. In this study, a feature importance-based dimensionality reduction method (FIA) is proposed to reduce the dimensionality while preserving the information of the feature space as much as possible. In the classification stage, this research reduces the computational load of the entire model and the computational pressure of the processor by grading.

## 2. Related Work

There have been many achievements in fall detection-related technologies. According to the mode of data perception, the detection of a fall can be divided into two methods: vision-based and wearable device-based.

The vision-based fall detection method uses a camera to collect the image information of a person. Under the influence of image processing technology and the algorithm, the movement state of the target character is obtained. Xu et al. [7] realized human fall detection based on depth information and the skeleton tracking technology of the Kinectv2 sensor. The algorithm first uses the depth information processing bone tracker to generate human joints, then it uses the BP neural network to recognize the posture and, on this basis, it detects falls. The neural network uses the data set generated by the Kinect tracker for training. In the test phase, different body trackers were used. The overall accuracy of the final body tracker was 98.5%, and the worst accuracy was 97.3%. Mansoor et al. [8] created a low-cost computing algorithm which relies on the head joints extracted by Kinect. This uses the standard deviation formula to check the changes in the head joint in the Y axis in each frame image. It compares the rate of change with the standard deviation of the experimental results, and then uses the KNN model for classification. In the test of 120 video samples with different activities, the algorithm achieved a 95% accuracy. Tao et al. [9] proposed an infrared sensor network system to analyze human behavior and detect falls in the home environment. In order to protect the user’s privacy, the short-term average of the response binary sequence is taken as the pixel value, and then the pixel value is trained as the feature. The average recognition rate reached 80.65% in eight activities of five subjects which were tested with the support vector machine. Vision-based fall detection methods are affected by many aspects, such as the location and number of cameras. At present, Kinect somatosensory devices are mainly used as image acquisition devices. Although this method will not add any burden to users themselves, it infringes the users’ privacy to a certain extent. In addition, the position of the camera is usually fixed, which means that it can only function in a fixed area. If we want to increase the detection area, we can only increase the number of cameras, which will undoubtedly increase the complexity and economic cost of the system.

The way that a fall is detected via wearable devices is through collecting data by wearing sensors and drawing conclusions after analyzing the data, or uploading the data to the upper computer for classification. Common fall detection algorithms include the threshold method and machine learning algorithm. Classification based on the threshold is a widely used method of fall detection technology, which detects falls by comparing the relationship between the sensor values and reference values. Quadros et al. [10] used a threshold-based method to classify behaviors, which achieved a 91.1% accuracy, 95.8% sensitivity, and 86.5% specificity in the experiment. In order to compare the differences between the two methods, Wu et al. [11] used random forest and threshold methods to classify behaviors. The former achieved an average accuracy of 94.9%, while the latter only achieved an accuracy of 91.6%. Ahn et al. [6] set the thresholds of a resultant acceleration, the angle, and angular velocity to gather the prediction of falls. In this paper, the public fall dataset Sisfall is used for testing. Because the threshold conditions are relatively loose, the accuracy rate is 90.3%, the sensitivity is 100%, and the highest specificity is 83.9%. The sensitivity has reached its peak, but the specificity can be further improved by refining the threshold conditions in the future. It is mentioned that the false alarm rate of lying down and sitting down is 100%, which indicates that the proposed method cannot distinguish between high acceleration behavior and falling. In order to provide an efficient fall detection, Yacchirema et al. [12] used the decision tree-based big data machine learning model running on the intelligent Internet of Things gateway to process and analyze the sensor readings. If a fall is detected, an alarm will be activated and the system will automatically react and send a notification to the group responsible for caring for the elderly. Ngu et al. [13] created a fall model based on the support vector machine and naive Bayesian machine learning algorithm. The experiment proves that the real positive rate of fall detection in the real world can be obtained with a 93.33% accuracy by adjusting the sampling frequency of stream data to calculate the acceleration characteristics on the sliding window and using the Naive Bayes machine learning model. Wang et al. [14] proposed a multi-source CNN integrated structure in the fall detection system, in which the data of the pressure sensor, acceleration sensor, and gyroscope are independently preprocessed and formatted. Then, the output characteristic maps of the three sensors are spliced into the overall characteristic map. In 800 falls and 1000 activities of daily living, the overall average accuracy of the fall detection system reached 99.30%, and the false positive rate was lower than 0.69%. Cahoolessur et al. [15] used the XGBoost algorithm and trained a machine learning model on the Sisfall dataset. Wearable devices worn at the waist use an accelerometer, microcontroller, global positioning system (GPS) module, and a buzzer design. The obtained acceleration data are converted into features and input into the machine learning model, and then the machine learning model will predict falls. Different from this literature, our proposed method is a hierarchical decision-making algorithm based on the XGBoost model. The algorithm is divided into three levels. The first level uses the threshold method to preliminarily evaluate the data and only transfers the fall type data to the next level. The second level is a classification algorithm based on XGBoost, which is used to analyze the remaining data types of the first level again. The third layer uses the comparison method to determine the falling direction. The threshold-based fall detection algorithm has the advantages of a low computation and fast detection speed, which is suitable for implementation on a simple microcontroller. The threshold method does not take into account the differences between the different populations and has a poor generalization ability. In contrast, the fall detection algorithm based on machine learning has a high sensitivity and specificity, and the generalization ability can also be guaranteed through parameter setting in the integrated learning model.

## 3. Materials and Methods

This section will elaborate on three aspects. First, this section will introduce the used dataset and describe the data preprocessing process. Second, this section will describe the dimensionality reduction method based on feature importance analysis. Finally, this chapter will introduce fall detection algorithms based on a hierarchical decision.

### 3.1. Data and Data Preprocessing

#### 3.1.1. Sisfall Dataset

This work uses the Sisfall dataset [16] in the training phase, which contains rich activities and a huge amount of data. Sisfall is captured using a device consisting of an accelerometer and a gyroscope. These included data on 19 activities of daily living (ADL) and 15 falls performed by 23 young adults, 15 ADLs performed by 14 participants over age 62, and data on all ADLs and falls performed by 1 participant aged 60 years. The detailed information is demonstrated in Table 1.

To determine the type of ADL and fall activity, this work surveys seniors living alone and nursing home administrators. The survey asks the elderly about the activities they were doing when they fell, the cause, direction, and impact site of the fall. The acquisition device is worn on the participant’s waist; the directions of the three axes of the sensor are shown in Figure 2; and the sampling rate is 200 Hz.

#### 3.1.2. Data Preprocessing

The data preprocessing stage includes four processes as shown in Figure 3, which are feature mapping, data sampling, feature extraction, and standardization. The sensor directly outputs an analog voltage, and then outputs a digital voltage value after passing through the analog-to-digital converter. The feature mapping stage maps the voltage value to the acceleration and angular velocity through a conversion formula. To facilitate the calculation, it is necessary to convert the hexadecimal data to decimal.

The raw samples of the dataset recorded the subjects’ acceleration and angular velocity data over a period of time. The width of the sample is 6, and the height is determined by the acquisition time and the sampling rate. The sampling rate is 200 HZ, and different types of behaviors have different sampling times. The specific time arrangement can be found in Table 1. During the preprocessing process, the width and height of the samples are constantly changing, the data sampling stage will reduce the height of the samples, and the feature extraction will further reduce the height of the samples to 1 row. In the first step, the sample width and height do not change, but the data unit and format are changed. Usually, normal, regular behavior predominates and falling behavior occurs less frequently. To study the fall behavior, it is necessary to increase the proportion of the fall data in the sample. This can be achieved by reducing the proportion of the ADL data in the sample, which is the work that needs to be done in the data sampling stage. After sampling, the width of the sample remains unchanged, and the height is consistent with the length of the set window. The length of the window in this paper is set to 60. Then, the resultant acceleration is obtained based on the 3-axis acceleration, and 44 time-domain features are extracted. After the first 3 steps of the calculation, the data can already be submitted to the model for training. However, the algorithm will focus on learning features with large values and ignore other features, so the data needs to be normalized in the end.

The data sampling pipeline. A sample contains data from ten seconds to hundreds of seconds, and the amount of data is very large. In order to reduce the amount of computation, the data can be reduced by sampling, and the most representative regions should be selected. The peak resultant acceleration is the maximum resultant acceleration within a period of time, and the impact moment can be determined by the time when the peak resultant acceleration occurs. For the sampling of the fall data, this work takes the impact moment as the center point of the sampling and then expands 30 sets of data to the left and right, which obtains 60 sets of sampled data. Each set of data represents the value of the triaxial acceleration and triaxial angular velocity at a certain moment. For the sampling of the ADL data, the measure in [17] is randomly selected within the time range. This will make the ADL data type richer, even including data that is not a part of the current behavior. In the process of collecting data, there is usually a preparatory period before the implementation of the behavior and a buffer period after the implementation of the behavior. As shown in Figure 4, segments T1 and T3 represent meaningless off-target behaviors. The T2 segment contains the peak resultant acceleration, which is the region that best represents the current behavior. If random sampling is used, it is difficult to select the T2 segment with a smaller proportion, and the final ADL sampling result will contain a large amount of data similar to the T1 segment and T3 segment. Walking and stilling result in a large number of similar ADL samples in the training set. As the number of valid samples in the training set decreases, the generalization ability of the model will decrease, making the actual test effect worse than the training effect. In this paper, the ADL data and the fall data are sampled in the same way, and the T2 segment in Figure 4 is the sampling area. This not only retains the most characteristic part of the ADL data, but also avoids obtaining similar sampling results for different ADL data.

The feature extraction pipeline. The essence of feature extraction is to integrate existing features to generate more useful features. Before formally extracting the features, this work first extracts the resultant acceleration based on the original features. Since the feature sampling stage uses the resultant acceleration as the center point, the resultant acceleration feature has been generated before the feature sampling is performed. The resultant acceleration is calculated from the three-axis acceleration, which is a non-negative value and represents the overall stress of the human body. Since the accelerometer does not remove the influence of gravity, the resultant acceleration and force here do not satisfy Newton’s second law. The calculation is:(1)$$ACCsvm=\sqrt{ACC_{x}^{2}+ACC_{y}^{2}+ACC_{z}^{2}}$$
where $ACC_{x}$, $ACC_{y}$, and $ACC_{z}$ represent the accelerations of the *X*-axis, *Y*-axis, and *Z*-axis.

This work studies the sampled data and extracts a large number of time-domain features by analyzing the feature of fall and ADL data [18]. Taking the acceleration as an example, the specific time domain feature and calculation methods are shown in Table 2.

Standardization transforms the data into a range with a mean of 0 and a variance of 1. The calculation formula is:(2)$$X=\frac{x-mean}{\sigma }$$
where mean is the mean of a feature in the dataset and σ is the standard deviation.

Normalization maps the data between 0 and 1, and the calculation formula is:(3)$$X=\frac{x-min}{max-min}$$

The mean and variance in standardization are calculated from all the data, and a small number of outliers will not affect the transformation results. The normalization process only involves the maximum and minimum values, and the rest of the values are not involved in the calculation, so outliers can easily affect the conversion results. If there is a range requirement for the data output result, or the data itself is relatively stable and there is no extreme maximum and minimum value, normalization is a better choice. However, in algorithms such as K-means, logistic regression, support vector machines, and neural networks, normalization is often the best choice. By comparing the characteristics of the two methods, this paper chooses the standardization method as the dimensionless method.

### 3.2. Feature Dimensionality Reduction

Dimensionality reduction aims to reduce the dimension of the feature space. The growth and update speed of datasets is accelerating, and the data are developing towards higher levels of high-dimensionality and unstructuredness [19]. Effective information is submerged in complex data, and the essential characteristics of the data are difficult to discover. How to achieve a low loss in the process of feature dimensionality reduction, maintain the properties of the original data, and obtain the optimal low-dimensional data has become an important goal of this paper.

#### 3.2.1. Dimensionality Reduction Based on Feature Importance

After the feature extraction, the feature space dimension is increased to 44 dimensions. In order to reduce the computational pressure of the processor, the dimension of the feature space must be reduced. There are two ways of obtaining a reduction in the feature dimensionality: feature extraction and feature selection. The former is to map the feature space to a space with smaller dimensions and more independent features, and its representative method is principal component analysis (PCA). The latter is to select a part of the features from the original feature space to form a new feature space. Its characteristic is that the feature itself does not change and still retains its original meaning, while PCA loses its original meaning after the reduction in the dimensionality.

In the feature selection method, how to measure the value of the feature becomes the primary problem of the feature selection. Feature importance is an indicator to measure the contribution of features to the prediction results, which can be used as a reference for the screening features. For feature importance, different algorithms have different calculation methods. XGBoost uses weight as the feature importance by default. There are three different indicators. The weight uses the number of times the feature splits nodes as an indicator. The gain uses the average gain of this feature split as an indicator. The cover uses the average coverage of this feature split as an indicator. The quotient of the score of each feature and the total feature score is the output as the feature importance:(4)$$importance=\frac{score}{\sum_{i=1}^{N}score_{i}}$$

Based on the feature importance, this paper designs a method to select the features. It aims to remove features that contribute less to a classification and improve the feature importance of all the features in the feature space. The main idea is to delete features through multiple iterations and continuously optimize the feature space to obtain the optimal feature space. The overall process is shown in Figure 5.

A lower threshold L is involved in the dimensionality reduction process, and there is no uniform standard for its value. It depends on the actual application scenario and is also closely related to the overall design of the system. This value is used to control the number of calculations in a single round of pruning, and to provide the theoretical minimum value of the feature dimension. In this paper, L is set to 10 to find the optimal feature space in a large range.

#### 3.2.2. Dimensionality Reduction Based on PCA

PCA is one of the most widely used feature dimensionality reduction methods, which aims to reduce the dimensionality of a dataset while preserving as much variability as possible [20,21]. In PCA, the features after the reduction in the dimensionality need to reflect the information contained in the original data as much as possible, and these features should be as independent as possible. Its specific approach is to map high-dimensional data into a low-dimensional space and replace the initial features with a smaller number of features. PCA extracts the most valuable information based on variance and ensures a linear independence between the features after the dimension reduction. Without the intervention of subjective parameters, although PCA is convenient for a general implementation, it also has the disadvantage that it cannot be personalized. At the same time, the use of PCA for dimensionality reduction needs to ensure that the data obeys the Gaussian distribution, otherwise the obtained main features may not be optimal.

### 3.3. Fall Detection Algorithm

The fall detection algorithm in this paper consists of three levels. The first level uses a threshold-based method to cull a large amount of ADL-type data to reduce the number of times the second level algorithm is used. The second level uses XGBoost to classify the activities judged by the first level as suspected falls. The third level identifies the direction of the fall. The overall process is shown in Figure 6.

#### 3.3.1. Fall Detection Based on Threshold

During the use of the fall detection system, the server is constantly receiving and classifying data. In order to reduce the number of times data are classified by XGBoost, this item sets a threshold level in front of XGBoost. Its function is to initially detect the type of data under the premise of a low computational complexity, and then pass it to XGBoost for classification when the behavior is difficult to distinguish using the threshold.

In this work, by carefully analyzing the features of fall and ADL data, it is found that the mean Y-axis acceleration $\mu_{ACCy}$ and the cumulative change $cum_{w}$ of the angular velocity amplitude can be used as the basis for the judgment. The data labels are set to ADL and FALL, and the data suspected of falling will be passed down for the next-step analysis. The purpose of threshold-based detection is to reduce the amount of non-fall data and retain fall data. Therefore, the threshold of each feature cannot be set too harshly, and a certain fault tolerance should be reserved for the suspected fall behavior. The judgment logic of the threshold method is shown in Figure 7.

#### 3.3.2. XGBoost Algorithm

XGBoost is composed of regression trees and the training method is similar to the gradient boosting decision tree (GBDT). Both of them train the next tree on the basis of one tree, and then calculate the difference between the output of the previous tree and the real value. By continuously training trees that bridge the gap, a combination of trees is finally used to simulate the real distribution. The training model usually defines an objective function ${\mathrm{obj}}^{\left(t\right)}$ and then optimizes the objective function in various ways to finally obtain the training model. The objective function is defined as follows:(5)$${\mathrm{obj}}^{\left(t\right)}=\overset{N}{\underset{i=1}{\sum }}L\left(y_{i},{\hat{y}}_{i}^{\left(t\right)}\right)+\overset{t}{\underset{j=1}{\sum }}\Omega \left(f_{j}\right)$$
(6)$$\Omega \left(f_{t}\right)=\gamma T+\frac{1}{2}\lambda \overset{T}{\underset{j=1}{\sum }}w_{j}^{2}$$

The objective function of GBDT consists only of the loss function, and the objective function of XGBoost adds a regular term part. The loss function describes the gap between the true value $y_{i}$ of the sample and the predicted value ${\hat{y}}_{i}^{\left(t\right)}$ and the sum of the loss values of all the samples represents the overall loss. The loss function also represents the degree to which the model fits the data. The larger the loss function, the lower the fitting accuracy, and the smaller the loss function, the higher the fitting accuracy. The complexity of the model can be controlled by the regular term $\Omega \left(f_{t}\right)$, and if the model complexity is too high, it will easily lead to overfitting. The larger the number T of leaf nodes, the greater the depth of the model. However, if it exceeds a certain limit, it is easy to cause data overfitting, and γ will be penalized for this situation. The larger the value of the leaf node, the larger the proportion of the regression tree in all the trees, which will also cause the problem of overfitting. At this time, λ will play the role of a penalty.

The regular term is the sum of the regular parts of the first t times, so the regular term in the objective function can be converted into the sum of the regular term of the first t−1 times and the regular term of the t-th time. When training the t-th tree, the first t−1 trees have already been trained. Therefore, the sum of the first t−1 regularization terms is known to the objective function, and this part does not affect the objective optimization. The optimization goals can be translated into:(7)$${\mathrm{obj}}^{\left(t\right)}=\overset{N}{\underset{i=1}{\sum }}L\left(y_{i},{\hat{y}}_{i}^{\left(t\right)}\right)+\gamma T+\frac{1}{2}\lambda \overset{T}{\underset{j=1}{\sum }}w_{j}^{2}$$

In machine learning, the general idea of optimizing an objective is to use gradient descent to find optimal parameters through multiple iterations. Since the tree model is a step function, it is not continuous in a mathematical expression, and its derivative cannot be obtained, so the gradient descent method cannot be used to optimize the objective function in XGBoost. By observing the structure of the tree, it is found that the loss value of each sample is only related to the current node value and has nothing to do with other node values. Therefore, the samples and nodes can be bound together and the minimum value of each node can be obtained, which can also minimize the overall goal in the end. Then, the optimization goal translates into:(8)$${\mathrm{obj}}^{\left(t\right)}|=\gamma T+\overset{T}{\underset{j=1}{\sum }}[\underset{i\in I_{j}}{\sum }L\left(y_{i^{\prime }}{\hat{y}}_{i}^{\left(t-1\right)}+w_{q\left(x_{i}\right)}\right)+\frac{1}{2}\lambda w_{j}^{2}]$$

When calculating the loss function by the node, the sample $x_{i}$ must belong to node j at this time. Therefore, $w_{q\left(x_{i}\right)}=w_{j}$, the optimization objective, can be transformed into the following formula:(9)$${\mathrm{obj}}^{\left(t\right)}=\gamma T+\overset{T}{\underset{j=1}{\sum }}[\underset{i\in I_{j}}{\sum }L\left(y_{i^{\prime }}{\hat{y}}_{i}^{\left(t-1\right)}+w_{j}\right)+\frac{1}{2}\lambda w_{j}^{2}]$$

In the process of the design of the algorithm, XGBoost adopts a series of optimization measures to improve the operation efficiency of the algorithm. When training a large dataset, the system cannot load the dataset into memory all at once, and the rest will be stored in chunks on different disks. Since the processor operates at a rate faster than the data read from the hard disk, the processor needs to wait for the data to be read into the memory after the processor completes the operation. Therefore, XGBoost opens a thread for reading data, so that the operation and reading are performed at the same time, reducing the waiting time of the processor. In addition, since the process of calculating the gain of all the features when building the tree does not affect each other, the training speed can be improved by parallelizing the processing.

The sampling of the features and eigenvalues effectively reduces the number of loops and the number of eigenvalue sorting. However, it is still necessary to sort the eigenvalues many times, which will take a lot of time. The strategy of XGBoost is to sort all the eigenvalues during the learning of the base learner and save the sorting results in the block structure, thus the subsequent process can directly obtain the sorting results from the block. Since only the first derivative and the second derivative are involved in the calculation of the gain, these two parameters are also added to the block structure. The block structure not only reduces the number of eigenvalue sorting but also improves the low cache hit rate, which effectively improves the training speed of XGBoost.

#### 3.3.3. Fall Direction Determination

This work divides the fall direction into forward, backward, to the left, and to the right. The direction can be classified by the threshold method or machine learning algorithm. For the threshold method, different individuals correspond to different thresholds, and the thresholds of some people cannot be used as a general standard. The scene of fuzzy judgment allows the algorithm to have a certain degree of fault tolerance, but it is no longer applicable in the scene of an accurate judgment. If the machine learning method is used, it can consider adding the label of the direction to the fall behavior and use the XGBoost model to judge the fall and the direction at the same time. The type of data translates to the ADL, fall forward, fall backward, fall to the left, and fall to the right. Since the number of ADL types in the dataset is much higher than the other 4 types, this will lead to imbalanced samples. After excluding the ADL data, there is little difference in the number of falls in the four directions. If the model to determine the fall direction is trained separately, the problem of a sample imbalance can be solved, but the small amount of data may lead to the overfitting of the model.

It can be seen from Figure 2 that the *Y*-axis is usually located in the vertical direction, and the *X*-axis and the *Z*-axis are located in the horizontal plane. After analyzing the fall process in combination with the sensor axis, it is found that a fall in any direction will lead to a change in the acceleration of the *Y*-axis, while the parameters of the *X*-axis and *Z*-axis vary with the direction of the fall. Through multiple fall experiments, it is found that the *Y*-axis acceleration does not show differences in the falls in all directions. Therefore, the fall direction cannot be determined by the acceleration of the *Y*-axis, while the horizontal plane parameters show obvious differences for falls in different directions. The initial state of the human body is standing still on the ground, and the *X*-axis and *Z*-axis accelerations are on the horizontal plane at this time. Additionally, it is not affected by gravity and other external forces, so the acceleration is 0*g*. The *Y*-axis acceleration is in the vertical direction and is not affected by other forces except gravity, so the acceleration value is 0*g*. When the human body falls forward, the *Z*-axis gradually turns from the horizontal plane to the vertical direction, and the gravitational effect on the body gradually increases, which causes the *Z*-axis acceleration to rapidly decrease from 0g to a negative number. When the human body falls to the right, as shown in Figure 8, the *X*-axis acceleration rapidly decreases from 0 to negative, and the amplitude reaches a peak value at time t1. If the human body falls backward or to the left, the acceleration increases rapidly from 0*g* to a positive number.

Therefore, the general direction of the fall can be obtained by comparing the magnitudes of the acceleration amplitudes of the *X*-axis and the *Z*-axis, and then the specific fall direction can be determined by the positive and negative signs. The specific implementation steps are shown in Figure 9.

## 4. Results

### 4.1. Evaluation Metrics

This work uses the accuracy, sensitivity, and specificity to evaluate the performance of the algorithm from various aspects, and the calculation process involves four parameters. *TP*: A fall occurs and is predicted to be a fall. *TN*: A fall does not occur and is predicted to be ADL. *FP*: A fall does not occur but is predicted to be a fall. *FN*: A fall occurs but is predicted to be ADL.

The accuracy refers to the proportion of correctly classified samples to the total number of classified samples:(10)$$Acc=\frac{TP+TN}{TP+TN+FP+FN}$$

The sensitivity is the ratio of correctly identified fall samples to all the fall samples:(11)$$Sen=\frac{TP}{TP+FN}$$

The specificity refers to the number of correctly identified ADL samples as a proportion of all the ADL samples:(12)$$Spe=\frac{TN}{TN+FP}$$

### 4.2. Result Analysis of Feature Dimensionality Reduction

After the feature extraction stage, the feature space dimension reaches 44 dimensions. This section first uses FIA to reduce the dimensionality of the feature space and uses XGBoost to classify the data during the dimensionality reduction process. Figure 10 shows the importance ratio and ranking of all the features in the initial feature space.

Point A represents the 12th most important feature and its importance ratio is 2.2%. Point B represents the 22nd most important feature and the feature importance ratio is 1.69%. The first half of the curve in the figure has a steep trend. With point A as the dividing point, the sum of the importance of the first 12 features accounts for 50% of the total. Taking point B as the dividing point, the sum of the importance of the first 22 features accounts for 80%, and the remaining half only accounts for 20%. These data show that most of the feature importance ratios are distributed in a small part of the features, and the rest of the features are unimportant or can be replaced.

Table 3 shows the distribution of feature importance ratios in each interval. It can be seen from the table that the importance of a large number of features accounts for between 1% and 2%, which is far from the highest point of 7.96%. However, as the number of features gradually decreases, the importance of some features will increase. On the one hand, as the total number of features decreases, the importance of each feature will increase slightly. On the other hand, some features can replace the positions of deleted features so that the proportion of their own importance increases.

The reduction in the feature dimensionality will be performed on the basis of the feature importance in the original feature space. Table 4 shows the process of dimensionality reduction and the results of each dimensionality reduction test. After the first reduction in the dimensionality, the feature space dimension is significantly reduced from 44 to 18 dimensions. After the second reduction in the dimensionality, the feature space dimension is reduced from 18 to 16. The first two dimensionality reductions are accompanied by small increases in the accuracy. When the third reduction in the dimensionality is performed, the dimension of the feature space decreases again, but it also leads to a decrease in the accuracy. Factoring in the performance of three dimensionality reductions, this paper selects the result after the second dimensionality reduction as the final feature space. When XGBoost uses this feature space, the accuracy of the test set reaches 97.14%, which is higher than 96.78% of the validation set, which indicates that using this feature space has a better generalization ability.

As the feature dimension decreases, the number of cycles experienced in the process of building the tree structure decreases synchronously, and the training time also decreases. To validate it, we test the time spent in training the model before and after the dimensionality reduction. The results show that it takes 0.47 s to train the XGBoost model before the dimensionality reduction and only 0.3 s after the dimensionality reduction, meaning a shortening of time by 0.17 s, and the test time is basically the same. These two experiments show that deleting some low-importance features can not only improve the accuracy of the model, but also reduce the training time of the model.

To compare the effect of the reduction in the dimensionality using PCA and FIA, this paper uses PCA to reduce the dimensionality of the features under the same conditions. The dimensionality reduction process is shown in Table 5, and the feature proportion represents the proportion of the feature information to be retained in the feature space after the reduction in the dimensionality.

As can be seen from Table 5, each time PCA is used for the dimensionality reduction is accompanied by a drop in the accuracy. In the process of reducing from 44 dimensions to 12 dimensions, the accuracy rate decreases slowly, and the accuracy rate is only reduced by 0.36% after three-dimension reductions. With a further dimensionality reduction, the accuracy begins to drop significantly. When using FIA to reduce the dimension to 18 dimensions, the accuracy rate is as high as 96.51%, and when it is reduced to 16 dimensions, the accuracy rate continues to increase to 96.78%. When using PCA to reduce the dimension to 18 dimensions, the accuracy rate is only 96.07%, which is 0.44% lower than the accuracy rate of FIA. Whether comparing the highest accuracy of the dimensionality reduction process or comparing the accuracy reduced to the same dimension, FIA always outperforms PCA.

Reference [22] proposed a splitting method for feature dimensionality reduction, which reduces one feature in the feature space at a time. Then, it compares the performance of the feature space with the best performance in this round of feature pruning and the current feature space. If the performance after deletion is better, use this feature space as the current feature space and repeat the deletion operation. If the performance before deletion is better, terminate the program, and the feature space before deletion is used as the optimal feature space. The principle of the splitting method is to exhaust all feature combinations and find the feature space with the best performance, and then perform an iterative operation. This method is simple in operation and clear in principle, but the object must be a low-dimensional feature space, because the exhaustive method requires a lot of computation. The dimensionality reduction method proposed in this work is guided by the feature importance, which reduces the number of attempts. Additionally, the splitting method does not achieve a completely exhaustive effect because it does not consider the interaction between the features, and the resulting features can only be locally optimal. If the feature combination is expanded, the optimal feature space can be found, but it will consume a lot of time.

### 4.3. Behavior Classification Based on Hierarchical Decision for Fall Detection

To test the performance of the fall detection algorithm on the optimal feature space, this paper collected 720 test samples, including 7 fall behaviors and 11 ADL behaviors. The test results are shown in Table 6. The results show that the fall detection algorithm can accurately distinguish ADL behaviors with a small amount of exercise, and behaviors with rapid acceleration changes such as jumping and jogging can also be basically correctly classified. Statistically, the fall detection algorithm tested 720 samples and obtained a 98.19% accuracy, 97.5% sensitivity, and 98.63% specificity.

This work extracts the data from the experiments with a clear fall direction. After further statistics and classification, the test results of the falling direction are obtained, as shown in Table 7.

Since the fall direction algorithm is at the third level, the errors of the first two layers of the algorithms are added to the error rate at the third level. This leads to a drop in the accuracy of the fall direction algorithm on the basis of a two-class classification.

### 4.4. Lead Time Statistics

This paper records the time when the fall is detected and the time when the impact occurs. The time difference between the time of the impact and the moment when the fall is detected is called the lead time. The presence of lead time means that the fall detection model can detect falls before the impact occurs. From Table 8, the lead time is relatively short, and the average lead time in the four directions is only 107.8 ms.

### 4.5. Literature Comparison

Table 9 summarizes the accuracy, sensitivity, and specificity of some of the literatures. What these papers have in common is that they all use the Sisfall dataset. Through comparison, it is found that this paper still has a certain room for improvement in terms of sensitivity, while the specificity and accuracy have a good performance.

## 5. Discussions

Different from traditional threshold-based detection techniques, this work uses a machine learning algorithm, which makes the fall detection model more adaptable and robust. Based on the analysis of the behavioral feature of falls and ADL, this work only uses the data of each sample near the peak resultant acceleration in the training phase, so as to ensure that the training data retains the behavioral features to the greatest extent. In the dimensionality reduction stage, PCA is usually used to reduce the dimension of the feature space, but this method has certain limitations. One is that the data after dimensionality reduction loses its original meaning, and the other is that the accuracy of the model usually decreases as the parameters decrease. Based on this, this work designs a dimensionality reduction method based on FIA. Despite the higher level of complexity introduced into the operation process, its effect is better than PCA.

Although this paper uses machine learning methods to detect falls, other parts are added in the process to improve the performance of the model. With the sampling rate of the sensor as high as 200 HZ, and the XGBoost frequently used for the data classification, the processor bears a greater computing pressure. To solve this problem, this work combines the characteristics of the threshold method and machine learning method. This work sets a threshold condition before the use of XGBoost. The purpose is to use the threshold for the preliminary data classification, so that XGBoost is only responsible for reclassifying the data of the fall type. With this threshold condition, most of the ADL data in daily life can be filtered out, and XGBoost only needs to focus on the fall behavior.

The fall detection algorithm proposed in this work performs well in both the Sisfall dataset and the self-collected data, and it can accurately judge the fall and the direction of the fall, but it still has several limitations. The fall data collected in this paper are all from young people, and the fall behavior is motivated by active will. In addition, anti-skid pads are also prepared to prevent the experimenters from being injured. These situations are quite different from the actual fall situation, and the test data are not completely consistent with the actual performance. In the future, the fall model is to be improved by expanding the database and adding real elderly fall data.

The performance of the algorithm in this paper also needs to be improved. Studies [23,24,25] show that falling will lead to significant changes in individuals’ physiological parameters, such as blood pressure and blood sugar. Therefore, blood pressure and blood oxygen can be added to the fall detection system to assist decision-making to improve the accuracy of the detection of a fall. In the experimental test, the accuracy of the two-class classification of the fall detection system reached 98.19%. However, simply detecting the fall behavior cannot reduce the injuries caused by the impact. When an elderly person falls, if an airbag device can be activated before the impact, the airbag can absorb the force generated by the impact, thereby reducing the risk of fall-related injuries [26].

Although the fall detection algorithm in this paper can detect falls about 100 ms before the impact, the lead time is too short to deploy the airbag. In the follow-up, this research will focus on extending the lead time as well as the airbag safety protection device, striving to achieve a complete fall detection and protection system.

## 6. Conclusions

According to the Seventh National Population Census in China, the country is in a moderately aging society and is on the verge of entering a deeply aging society. As the proportion of those who are elderly increases, there will be more and more problems related to the elderly, and falls are one of the most typical problems. Once a fall occurs, the elderly person will suffer both physical and psychological pressure, which will also increase the financial burden on their families. In many cases, the elderly will not be found in the first place after falling because many are likely be alone at home. With the passage of time, the injury will worsen and the best treatment time will be missed, resulting in sequelae. Therefore, the timely detection of falls will have a positive effect on protecting the life and property of the elderly and their families. The fall detection algorithm will play this important role.

This work uses inertial sensors placed on the waist of the experimenter to collect motion data, and then develops a fall detection model based on the XGBoost algorithm. The acceleration and angular velocity data of the human body are collected by the inertial sensor. According to the characteristics of the daily behavior and fall behavior, this study extracted 44 commonly used features. Given the large number of features, this will increase the computational pressure on the processor and further increase the time the model spends in the training phase. This work adopts an FIA-based strategy to reduce the number of features, which reduces the feature space from 44 dimensions to 16 dimensions. Experiments show that the FIA-based dimensionality reduction method can greatly reduce the dimension of features and reduce the time spent in the training phase. In addition, it can also slightly improve the accuracy of the model. The comparative experiments based on FIA and PCA also show that the performance of the FIA method proposed in this paper is better than that of the PCA method. This study adopts the idea of grading in the classification stage. The threshold method is used to initially classify the data at the first level; the XGBoost algorithm is used to accurately classify the data at the second level; and finally the fall direction is determined at the third level. This method greatly reduces the frequency of the usage of XGBoost and the computational load of the processor. On the basis of using FIA for the reduction in the feature dimension and hierarchical decision-making for identification, the accuracy of judging falls in this paper reaches 98.19%; the sensitivity reaches 97.5%; and the specificity reaches 98.63%. The experimental results show that the hierarchical decision fall detection algorithm proposed in this paper can be used to detect falls and fall directions.

## Figures and Tables

**Figure 1 sensors-23-00107-f001:**
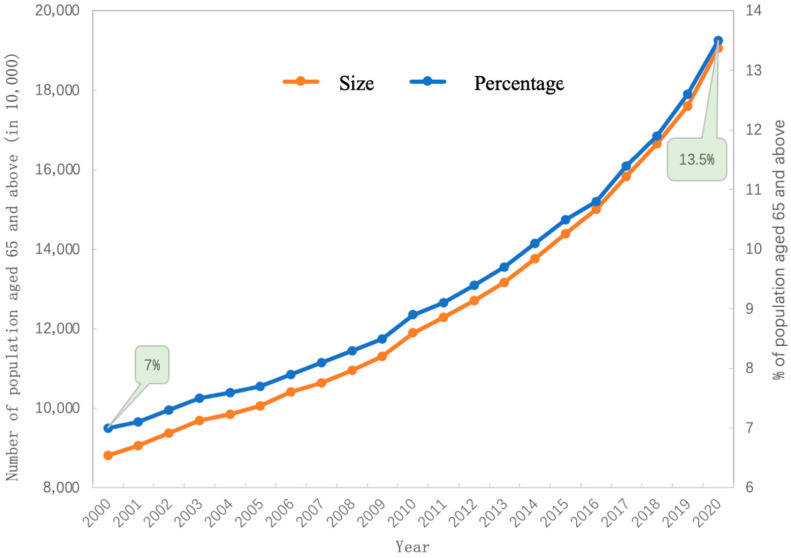
Trend of population aging in China.

**Figure 2 sensors-23-00107-f002:**
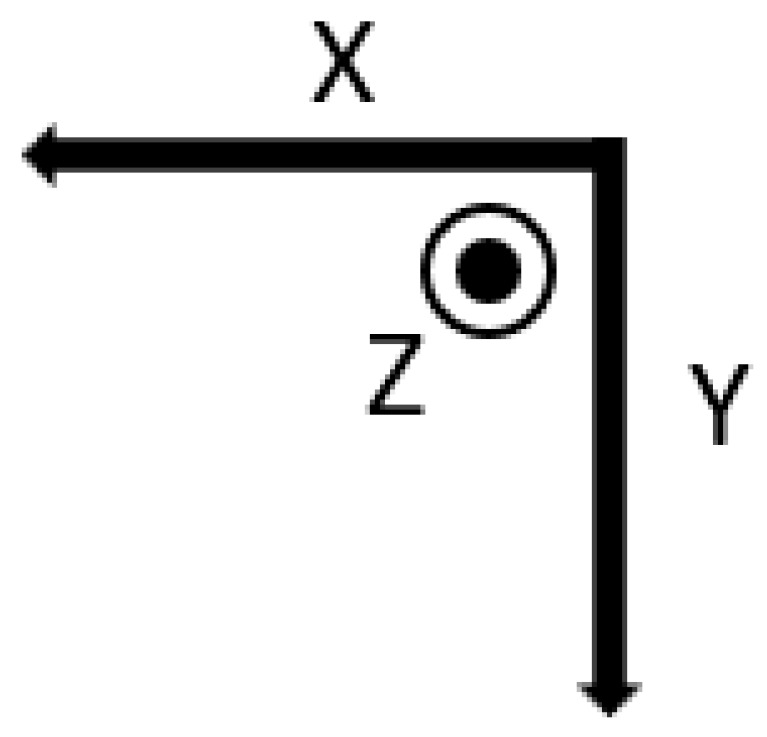
Axial direction of the sensor.

**Figure 3 sensors-23-00107-f003:**
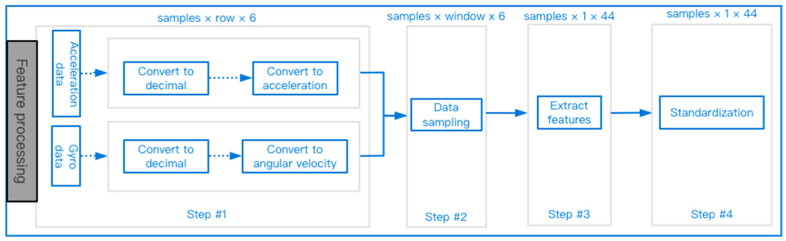
Data preprocessing process.

**Figure 4 sensors-23-00107-f004:**
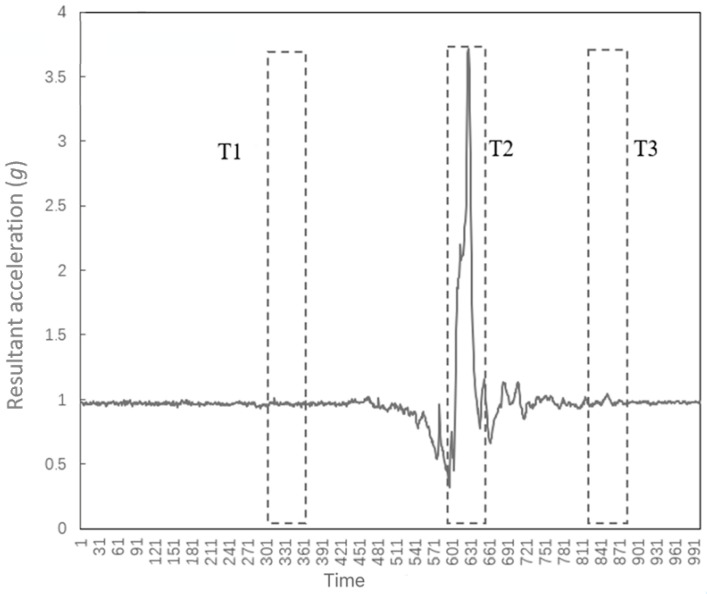
Position of the window.

**Figure 5 sensors-23-00107-f005:**
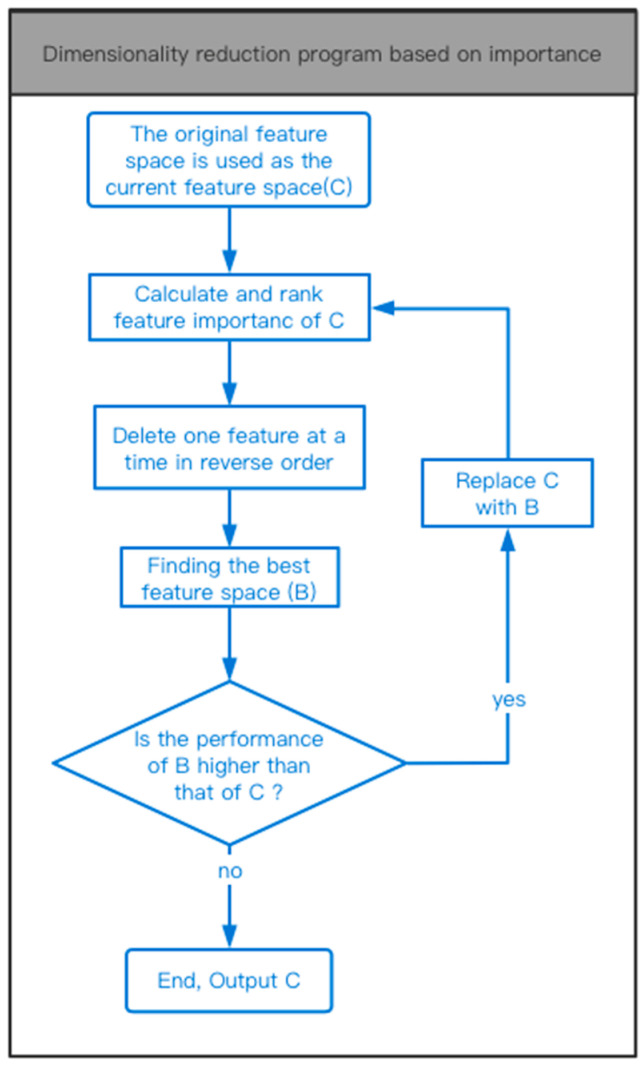
The process of dimensionality reduction.

**Figure 6 sensors-23-00107-f006:**
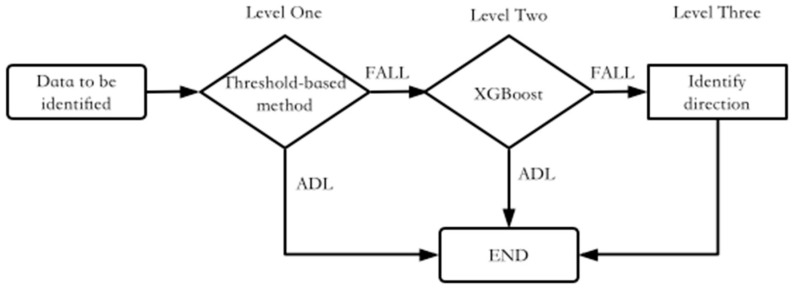
The model of fall detection.

**Figure 7 sensors-23-00107-f007:**
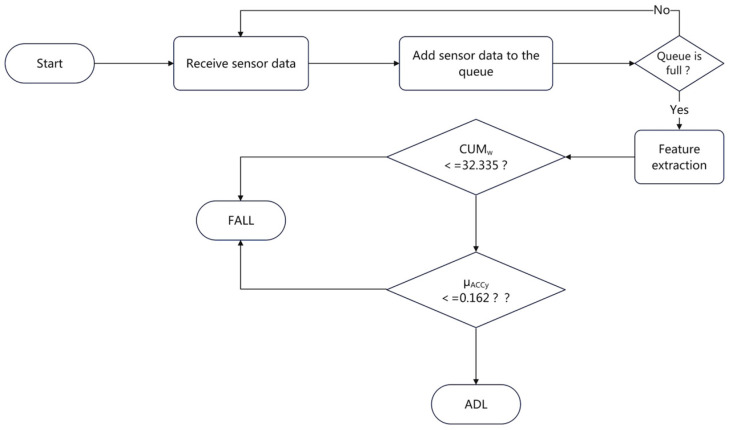
Frame diagram of threshold method.

**Figure 8 sensors-23-00107-f008:**
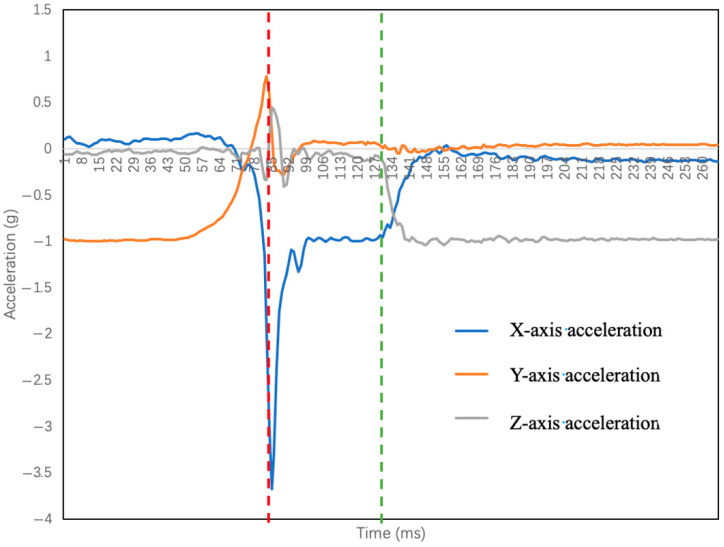
Fall to the right.

**Figure 9 sensors-23-00107-f009:**
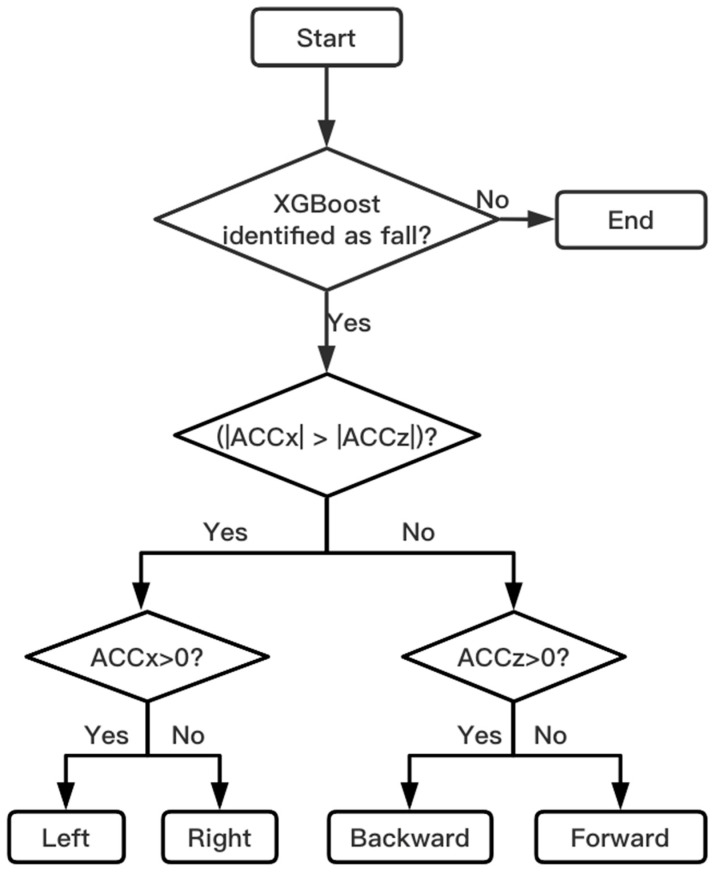
Process for identifying fall direction.

**Figure 10 sensors-23-00107-f010:**
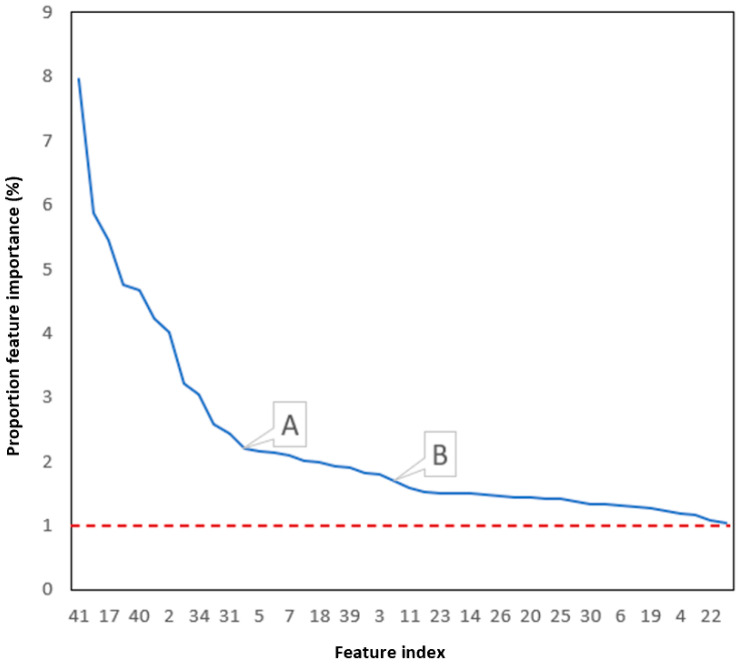
Proportion of importance of original feature space.

**Table 1 sensors-23-00107-t001:** Types of activities of daily living and fall.

Code	Activity	Trials	Duration
D01	Walking slowly	1	100 s
D02	Walking quickly	1	100 s
D03	Jogging slowly	1	100 s
D04	Jogging quickly	1	100 s
D05	Walking upstairs and downstairs slowly	5	25 s
D06	Walking upstairs and downstairs quickly	5	25 s
D07	Slowly sit in a half height chair, wait a moment, and up slowly	5	12 s
D08	Quickly sit in a half height chair, wait a moment, and up quickly	5	12 s
D09	Slowly sit in a low height chair, wait a moment, and up slowly	5	12 s
D10	Quickly sit in a low height chair, wait a moment, and up quickly	5	12 s
D11	Sitting a moment, trying to get up, and collapse into a chair	5	12 s
D12	Sitting a moment, lying slowly, wait a moment, and sit again	5	12 s
D13	Sitting a moment, lying quickly, wait a moment, and sit again	5	12 s
D14	Supine to side, and change to one’s back	5	12 s
D15	Standing, slowly bending at knees, and getting up	5	12 s
D16	Standing, slowly bending without bending knees, and getting up	5	12 s
D17	Standing, get into a car, remain seated and get out of the car	5	25 s
D18	Stumble while walking	5	12 s
D19	Gently jump without falling	5	12 s
F01	Fall forward while walking caused by a slip	5	15 s
F02	Fall backward while walking caused by a slip	5	15 s
F03	Lateral fall while walking caused by a slip	5	15 s
F04	Fall forward while walking caused by a trip	5	15 s
F05	Fall forward while jogging caused by a trip	5	15 s
F06	Vertical fall while walking caused by fainting	5	15 s
F07	Fall while walking, with use of hands in a table to dampen fall	5	15 s
F08	Fall forward when trying to get up	5	15 s
F09	Lateral fall when trying to get up	5	15 s
F10	Fall forward when trying to sit down	5	15 s
F11	Fall backward when trying to sit down	5	15 s
F12	Lateral fall when trying to sit down	5	15 s
F13	Fall forward while sitting, caused by fainting or falling asleep	5	15 s
F14	Fall backward while sitting, caused by fainting or falling asleep	5	15 s
F15	Lateral fall while sitting, caused by fainting or falling asleep	5	15 s

**Table 2 sensors-23-00107-t002:** Time domain features.

Name	Formula
Mean value	$mean=\frac{1}{n}\overset{n}{\underset{i=1}{\sum }}a_{i}$
Maximum value	$max=max\left\{x_{i}|i\in \left\{1,2,3\ldots n\right\}\right\}$
Minimum value	$min=min\left\{x_{i}|i\in \left\{1,2,3\ldots n\right\}\right\}$
Standard deviation	$std_{t}=\sqrt{\frac{1}{n}\overset{n}{\underset{i=1}{\sum }}{\left(a_{i}-mean\right)}^{2}}$
Value range	range=max−min
Variation	Δa=a(t+T)−a(t)
Cumulative change	$cum=\int_{0}^{T}adt$
Average change rate	$\frac{1}{T}\int_{0}^{T}\left|a^{\prime }\right|dt$
Signal amplitude area	$SMA=\left(\sum_{i=1}^{n}\left(\left|a_{xi}\right|+\left|a_{yi}\right|+\left|a_{zi}\right|\right)dt\right)/T$

**Table 3 sensors-23-00107-t003:** The distribution of feature importance.

Range	Quantity
[0, 1)	0
[1, 2)	28
[2, 3)	7
[3, 8)	9

**Table 4 sensors-23-00107-t004:** The process of feature dimensionality reduction.

Number	Feature Space Dimension	Accuracy on Validation Set (%)
0	44	96.31
1	18	96.51
2	16	96.78
3	15	96.44

**Table 5 sensors-23-00107-t005:** The process of dimensionality reduction using PCA.

Feature Proportion	Feature Dimension	Accuracy on Validation Set (%)
1	44	96.31
0.99	29	96.19
0.95	18	96.07
0.9	12	95.95
0.85	9	95.24
0.8	7	91.91

**Table 6 sensors-23-00107-t006:** Results of behavioral tests.

Test Behavior	Total Samples	Fall	ADL
Walk	40	0	40
Jogging	40	1	39
Jump	40	2	38
Turn around	40	0	40
Go upstairs	40	0	40
Go downstairs	40	0	40
Stand up after sitting down	40	0	40
Sit in a chair and try and fail	40	3	37
Stand for a while, bend knees, stand up	40	0	40
Stand for a while, squat down to tie shoelaces, stand up	40	0	40
Stand for a while, bend over, stand up	40	0	40
Leaning forward due to slipping while walking	40	39	1
Leaning backwards due to slipping while walking	40	40	0
Leaning to the left due to slipping while walking	40	38	2
Leaning to the right due to slipping while walking	40	39	1
Leaning forward due to tripping	40	39	1
Falling backwards when trying to sit	40	40	0
Fall sideways while sitting	40	38	2

**Table 7 sensors-23-00107-t007:** Test results for the direction of fall.

Experimental Content	Total	Forward	Backward	Left	Right	ADL	Acc (%)
Fall forward	80	76	0	2	1	1	95
Fall backward	80	0	79	0	1	0	98.75
Fall to the left	40	0	2	36	0	2	90
Fall to the right	40	1	2	0	36	1	90

**Table 8 sensors-23-00107-t008:** Test results for the lead time of fall.

Experimental Content	Total	Average Lead Time (ms)
Fall forward	80	102.5
Fall backward	80	130.8
Fall to the left	40	83.7
Fall to the right	40	114.2

**Table 9 sensors-23-00107-t009:** Test results of different literatures.

Method	Acc (%)	Sen (%)	Spe (%)
Literature [6]	90.3	100.0	83.9
Literature [12]	91.0	100.0	94.0
Literature [13]	93.0	94.0	92.0
Literature [15]	96.0	97.0	93.0
Ours (Sisfall)	97.14	97.24	97.03
Ours (self-made dataset)	98.19	97.5	98.63

## Data Availability

The data presented in this study are available on request from the corresponding author. The data are not publicly available due to ongoing research.

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
