# Peer review of "Fall Detection Algorithm Based on Inertial Sensor and Hierarchical Decision"

_sensors, 2022, doi:10.3390/s23010107_

Round 1

Reviewer 1 Report

The problem of fall detection globally affects the lives of the elderly. It is an interesting approach proposed by the authors, however, I find that the same approach has been published previously (http://www.scielo.org.za/scielo.php?script=sci_arttext&pid=S2309-89882020000100003) XGboost applied to the Sisfall dataset. 

This was not covered in the literature review. The authors would need a strong motivation as to how their approach adds novelty to what has already been presented.

Two other points of contention: the introduction dedicates too much text to describe the problem and setting the context. The literature review is close to non-existant and only covers a single source ([6]) in any detail.

The results are impressive but there is a lack of context specifically in table 9. The reader has no idea what type of methods the literature for comparison used and thus if it is a fair comparison. 

Author Response

Thank you for your valuable suggestions. Please see the attachment for my response.

Reviewer 2 Report

The authors design a fall-detection system by integrating several techniques, such as dimensionality reduction and hierarchical decision-making. The experimental results demonstrate the system’s superiority in terms of accuracy and responsiveness.

However, there are some drawbacks that might be noticed.

1. Although the authors compared their work with others, the comparison mainly focuses on non-ML based solutions. The authors are encouraged to give a brief introduction to the existing ML-based systems, describing their weaknesses and further compare the performace among them.

2. As this paper considers data from wearable devices, the power consumption should be a significant factor to be measure. Adding such a benchmark would give the reader a more comprehensive understanding of such methods.

3. The paper needs a careful revision. For example, the y-axes of figures are inconsistent.

Author Response

Thank you for your valuable suggestions, and my reply is as follows. See the appendix for the revised paper.

Point 1: Although the authors compared their work with others, the comparison mainly focuses on non-ML based solutions. The authors are encouraged to give a brief introduction to the existing ML-based systems, describing their weaknesses and further compare the performance among them.

Response 1: Thank you very much for your constructive comments. We have added a new Related Work chapter to conduct the discussion for relevant research. In addition, in Table 9, in order to make the experiment more comparative, we readjusted the compared methods, the new methods we adopt in Table 9 are machine learning based methods, and these methods are described in Related Work as follows:

Ahn et al. [6] set the thresholds of resultant acceleration, angle and angular velocity to realize the prediction of falls. In this paper, the public fall dataset Sisfall is used for testing. Because the threshold conditions are relatively loose, the accuracy rate is 90.3%, the sensitivity is 100%, and the highest specificity is 83.9%. The sensitivity has reached the peak, and the specificity can be further improved by refining the threshold conditions in the future. It is mentioned that the false alarm rate of lying down and sitting down is 100%, which indicates that the proposed method cannot distinguish between high acceleration behavior and falling. In order to provide efficient fall detection, Yacchirema et al. [12] uses the decision tree based big data machine learning model running on the intelligent Internet of Things gateway to process and analyze sensor readings. If a fall is detected, an alarm will be activated and the system will automatically react and send a notification to the group responsible for caring for the elderly. Ngu et al. [13] created a fall model based on support vector machine and naive Bayesian machine learning algorithm. The experiment proves that the real positive rate of fall detection in the real world can be obtained with 93.33% accuracy by adjusting the sampling frequency of stream data to calculate the acceleration characteristics on the sliding window and using Naive Bayes machine learning model. Cahoolessur et al. [15] used XGBoost algorithm and trained machine learning model on Sisfall dataset. Wearable devices worn at the waist use accelerometer, microcontroller, global positioning system (GPS) module and buzzer design. The obtained acceleration data are converted into features and input into the machine learning model, and then the machine learning model will predict falls.

Point 2: As this paper considers data from wearable devices, the power consumption should be a significant factor to be measure. Adding such a benchmark would give the reader a more comprehensive understanding of such methods.

Response 2: Thank you very much for your constructive comments. In this paper, the hardware equipment we use are mature overall accessories. The main goal of this paper is to design a more efficient fall detection algorithm, and power consumption is a secondary issue we consider. And we also conducted a brief test, and the hardware power consumption is within the acceptable range.

Point 3: The paper needs a careful revision. For example, the y-axes of figures are inconsistent.

Response 3: Thank you very much for your constructive comments. We have adjusted these figures accordingly, and hope they can meet your requirements.

Round 2

Reviewer 1 Report

The updates have sufficiently addressed my concerns. My only outstanding issue is grammatical/language errors. 

Reviewer 2 Report

The authors have made a fair amount of changes to improve the paper and have answered the questions I put forward.